# Safety Performance Assessment of Construction Sites under the Influence of Psychological Factors: An Analysis Based on the Extension Cloud Model

**DOI:** 10.3390/ijerph192215378

**Published:** 2022-11-21

**Authors:** Junlong Peng, Qi Zhang

**Affiliations:** School of Traffic & Transportation Engineering, Changsha University of Science and Technology, Changsha 410114, China

**Keywords:** construction safety, extension cloud model, psychological hazards, leader–member exchange ambivalence, safety evaluation, safety performance

## Abstract

Psychological hazards within organizational structures of construction sites are difficult to detect and can have significant negative impacts on safety performances when such hazards erupt. At present, most safety performance assessment models for construction sites ignore psychological factors. Therefore, in order to reveal psychological hazards within construction site organizations and to avoid damage caused by psychological hazards to safety performances, this paper evaluates the safety performances of construction sites by focusing on leader–member exchange ambivalence as the main trigger point. The evaluation system and evaluation criteria are established through three aspects: building scale, emotional orientation, and stability factors. The hierarchical analysis method, game theory, and extension cloud model are combined to make evaluation results more objective and credible. Moreover, a construction project with high technical requirements, high investment, and complex construction conditions (defined as a complex project) and an ordinary construction project with low technical difficulty and simple construction conditions (defined as a general project) were selected for analysis. The evaluation results indicate that both complex projects and general projects have safety hazards regarding psychological orientations. Finally, this paper makes some suggestions from three aspects: management system and corporate culture, building site intelligence, and social opinion to improve the safety performances of construction sites. The evaluation results are the same as actual operation results, which verify that models proposed in this paper can be used for safety performance evaluations of actual construction projects and provide help for managers to grasp overall safety levels.

## 1. Introduction

For a long time, project managers, as the main leaders of construction processes, were considered to be the main ones responsible for the safety performances of construction projects [1]. However, the construction industry’s accident reports reveal certain flaws in this model. On the one hand, with increases in the professionalism of participants and further improvement of supervision mechanisms, contradictions between parties that involve construction have gradually come to the forefront, and the dominant position of the construction side has been subject to much dissatisfaction [2]. On the other hand, construction site operators have a low sense of organizational participation, low organizational status, and rely more on organizational coordination of leaders for collaborative work of different working groups [3]. This requires a close cooperative relationship between various leaders. In such a situation, the ambiguities of hierarchical relationships between the party leaderships arise, which leads to the inability of staff in construction sites to accurately judge their own organizational statuses and intimate relationships with leaders [4]. Therefore, the following should be addressed: avoiding the psychological orientation of staff members in construction sites (affecting the synergy and cooperation of work branches), exploring and analyzing the factors affecting the psychological states of staff, maintaining quality safety performances in construction project sites, and evaluating the overall psychological safety situations.

In many studies, collaborative capabilities have proven to be essential to achieving goals [5,6,7], demonstrating that increased mutual understanding and acceptance of employees lead to better safety performances. However, during relatively short durations in construction projects, parties involved in the construction cannot quickly reach a cooperative understanding, and it is common for staff to be separated from each other [8]. Under such circumstances, managing construction sites and company culture are key factors in ensuring safety performances and play important roles in promoting personnel collaboration [9]. Efficient personnel collaboration communication requires a clear perception of organizational status and hierarchical relationships between all parties involved in construction. Based on timely information communication, operators can dynamically adjust their work strategies according to the project’s requirements, thus achieving precise control of the project’s goals and responding rapidly when problems are encountered. However, the organizational relationships within construction sites are ambiguous and complex and the organizational atmosphere and management systems of party leaderships are different.

A comprehensive and clear construction site safety assessment is particularly important in complex environments. Safety assessment in the construction industry is not uncommon [10,11,12]. Previous studies have focused on the mechanisms of the interactions between objective factors, including real and social environments in which construction sites are located, the objective states of the operators (professional skills, physical states, etc.), supervision efforts, safety systems, financial investments, etc. However, to comprehensively and accurately assess the safety performances of construction sites, previous studies have mostly conducted static evaluations from overall perspectives, lacking an in-depth understanding of the dynamic relationships between personnel, and not focusing on the deviation of safety performances under the influence of employees’ psychological states. However, from the perspective of macro-management mechanisms in construction projects, negative interpersonal relationships can also be hidden under good cooperative performances [13]; a lack of information leads to safety hazards that cannot be restrained timely and effectively. Therefore, a good path to restrain psychological hazards must be explored.

### 1.1. Leader–Member Exchange Ambivalence in Construction Site

In the intersection of architecture with psychology and management sciences, the term leader–member exchange refers to the reciprocal process of communication and collaboration between leaders and subordinates, which usually has a greater impact on employees [14]. In the context of leader–member exchange, leaders often make highly conflicting demands on employees to effectively achieve organizational goals, while providing adequate support to the employees. The demanding–caring behavior of leaders can easily lead employees to hold ambivalent evaluation perceptions about subordinate relationships. Ambivalent experiences refer to people’s perceptions about the positive and negative evaluations of things, which generally lead to negative consequences [15]. Workplace relationships are critical sources of ambivalent experiences because of the long-term continuity, high frequency of interactions, and diversity [16].

Compared to the general industry, the construction industry is characterized by a large number of people involved as well as an ad hoc organizational structure, resulting in a more complex environment [17]. The leader–member exchange ambivalence also applies to the staff on a construction site. During a construction period, most of the staff live on the construction site, which gives them a lot of time to become acquainted with each other. Moreover, this is difficult to avoid. When subordinates experience good intentions from their superiors, they may increase their work commitments and accomplish project goals more efficiently. He et al. [18] proposed that the influence and contribution of the leader–member exchange relationship is positively related to safety behaviors and that the influence and professional respect of the leader–member exchange relationship have indirect effects on safety involvement through communication competence, demonstrating a mediating role of communication competence and construction safety. Liu et al. [19] suggested that the leader–member exchange relationship can act as a mediator and that a good leader–member exchange relationship can reduce workplace deviance. Abu et al. [20] proposed that the quality of the leader–member exchange relationship can mediate group-oriented behavioral confrontations and that the leader–member exchange relationship can, directly and indirectly, influence positive intergroup relationships.

However, frequent interactions create ambiguity in harsh hierarchical relationships [21], providing conditions for inducing ambivalence in employees. In the process of establishing relationships between superiors and subordinates, leaders need to maintain a certain hierarchical distance from their employees due to the limitations of the organizational hierarchy; they are also attempting to establish closer interpersonal relationships [22]. The damaging effects surrounding ambivalence from employees have been widely demonstrated. Han [23] proposed that leader–member exchange ambivalence has a negative impact on the well-being of employees, as evidenced by lower work commitments and increased emotional exhaustion. Using two surveys involving 387 employees and 110 supervisors, Huang et al. [24] suggested that leader–member exchange conflicts exacerbate job anxieties in employees. Lee et al. [25] found that leader–member exchange ambivalence has a negative effect on employee task performances. van Harreveld et al. [15] argued that leader–member exchange ambivalence is usually accompanied by uncertainty, and to eliminate this discomfort, employees usually adopt defensive coping strategies.

### 1.2. Contributions

To solve the problem of invisible psychological conflicts and ambiguous psychological perceptions of staff at construction sites, we explored a safety performance evaluation model under the influence of psychological states. Rooted in interactions of management levels within construction sites, management levels at construction sites are divided into three parts (investor, constructor, and subcontracted labor); an extension cloud model is introduced to rate the index system hierarchically. This model was applied to two actual construction projects to make this study more reasonable and universal. The main contributions of this paper are as follows:

(1) Beginning with an ambivalent relationship between management groups, we explored the influence of ambiguous superior–subordinate intimacy on work engagement and the psychological perceptions of staff in construction sites.

(2) An index system for evaluating the safety performances of building sites under the influence of psychological states was established, and quantitative indicators of ambivalent experiences among staff members were proposed.

(3) We verified feasible paths of constraining implicit psychological hazards affecting the safety performances of building sites, found key factors, and propose suggestions to suppress the hazards.

### 1.3. Text Structure

The paper is structured as follows: The Section 2 introduces the research methodology, which mainly includes: research background, establishment of evaluation index system, the weight assignment, and safety level determination. The Section 3 presents an evaluation model proposed in two practical cases. The Section 4 and Section 5 present the discussion and conclusion, including an analysis of the evaluation results.

## 2. Research Methodology

### 2.1. Description of Safety Assessment Framework

The leader–member exchange ambivalence is experienced throughout construction projects and is reflected at each level of leadership. In this paper, the management of a construction project is partitioned into three levels—management of investors, management of constructors, and management of labor subcontractors. The investors refer to actual holders of construction projects after completion; they have absolute control over the construction projects. Investors’ management mainly includes the construction project management of the investment company and supervision unit that assists in the management. The constructors refer to the actual leading organization in the project’s construction process, which is formed temporarily and mainly appointed by a construction contractor; the leadership of the construction side mainly includes the top organizer (e.g., the project manager or chief technical person) who directs the construction process and the person in charge of all aspects of the construction site (construction technicians, safety supervisor, etc.); labor subcontractors refer to various types of skilled labor personnel that exist in the construction site; management of labor subcontractors mainly includes various types of skilled laborers. Management of the labor subcontractor mainly includes foremen, intermediaries of technical outsourcing, etc.

A perfect construction project requires participation from several specialized agencies; in addition to the three parties proposed above, there is also the participation of survey units, design units, rental units, suppliers of various materials, etc. However, during the construction process, these units cannot stay on the construction site for a long time, and they do not have much influence from the point of view of the leader–member exchange ambivalence affecting the safety performance of the construction site. Therefore, these units are not included in this paper.

In general, investors, as ultimate controllers of projects, have the highest leadership, but their expertise is weak; the investors may be unable to give comprehensive and accurate guidance by relying more on the advice from consultants and supervisors who manage the project. It is common that the expertise of investors is not commensurate with the power they hold [26]. The result of this phenomenon is that the subjective belief from construction and labor subcontractors, i.e., that they will satisfy investors, often produces very different results.

The constructors, as actual leaders of construction processes of construction projects, are extremely reliable in their expertise but are subject to the management of investors in every way [27]. When investors rely too much on supervisors, constructors even need to show very ambiguous attitudes toward supervisors. In this case, unclear superior–subordinate relationships between constructors and supervisors can be conflicting. On the one hand, they need to be on the same side as supervisors to obtain support for their work; on the other hand, they are afraid of not being able to communicate their true attitudes to investors through supervisors, thus creating negative work input.

For a long time, labor subcontractors have been the most indispensable party in construction projects, but they have to follow the direction of constructors and investors in terms of project decision-making. With the spread of education for all in China and the standardization of the construction industry, the quality of China’s construction workers has further improved and their individuality and construction involvement have been met [28], giving them a place in project decision-making.

The tripartite relationship is complex and intertwined with management styles. The tripartite relationship is depicted in Figure 1. The blurring of superior–subordinate relationships and mismatches between expertise and executive authority both deepen the intensity of the ambivalence generated. The results of ambivalent experiences are often negative, especially in the area of safety performance, which is the most worrisome aspect of construction projects. For example, constructors fear that the construction process will be deliberately “nitpicked” by a supervisor, so they deliberately avoid the supervisors in some sensitive construction parts and work quickly, which causes safety hazards. In addition, some workers, relying on their good relationships with constructors, do not comply with the safety systems and disrupt the safety atmosphere, which results in serious safety accidents.

A clear organizational structure with well-defined rules and regulations is one of the effective means to reduce the leader–member exchange ambivalence [29], but this ambivalence is difficult to erase completely. The leader–member exchange ambivalence is an implicit factor that cannot be precisely described using a single indicator. In order to more precisely resolve the uncertainty and ambiguity of each party’s psychological state, this paper establishes an evaluation system to profile the leader–member exchange ambivalence through some easily accessible explicit indicators. This paper proposes a cloud model-based conventional safety evaluation method under the influence of the psychological state of the construction site based on the cloud model theory of qualitative and quantitative uncertainty transformation.

The research methodological framework of this paper is as follows: (1) Establishment of the safety evaluation index system. (2) Allocation of safety evaluation index weights. (3) Comprehensive analysis of the extension cloud model. (4) Example analysis. (5) Safety evaluation conclusion. The specific research method framework is shown in Figure 2.

### 2.2. Safety Evaluation Index System

The situations of construction sites are extremely complex; coordinated operations involving multiple heavy types of machinery, construction tools, and a large number of workers endow leaders with great management difficulties, but states exhibited by different types of construction sites can vary greatly, and situations of construction sites are the results of interactions between multiple factors. Therefore, analyzing the influential indicators affecting the management decisions of leadership in construction sites and exploring reasonable key factors are prerequisites for establishing evaluation systems featuring safety performances of construction projects under the influence of psychological states.

The establishment of an evaluation index system, a specific analysis for specific problems, and a combination of quantitative and qualitative analyses need to be targeted. The selection of indicators also adopts the idea of a hierarchical analysis to rank problems and decompose the goals of the safety performances of construction projects under the influences of psychological states. In this paper, the evaluation system of the safety performance of a construction project under the influence of a psychological state was constructed from three perspectives—building scale, emotional orientation, and stability factors—and was decomposed into eighteen specific indicators. These indicators were extracted through relevant Chinese construction industry codes, existing literature, expert discussions, two project managers, three senior engineers, two experts engaged in research in this field, and three construction technicians who were invited for collaborative discussions. The Delphi method was used to invite scholars or experts in the field to determine and evaluate the indicators to be used. The information on the participants is shown in Table 1. The facilitator summarized the report and discussed the applicability of these assessment indicators. Afterward, the facilitator provided feedback to the experts and repeated this operation until an agreement was reached. The evaluation index system is shown in Table 2. Specific indicators are analyzed as follows.

(1) Building scale:

The scale of the construction project is the basis for all operations within the project. As the scale increases, so do the influencing factors that ensure the proper functioning of the construction projects. In addition to the basic conditions of the number of participants ($B_{3}$), amount of investment ($B_{1}$), floor area ($B_{2}$), and built environment ($B_{4}$), the factor of social opinion ($B_{5}$) received more attention after the outbreak of COVID-19 [30]. Now, with an explosion of information, information about a construction project that receives attention is quickly disseminated to all parts of society via the internet, such as the construction processes of Huoshenshan and Leishenshan hospitals [31,32]. Social opinions magnify the behaviors of employees, and operations that conform to high-quality specifications will be praised by the public; the opposite will be indelibly shamed. There is also a great deal of psychological pressure on employees to be more compliant with the rules [33]. Many studies have demonstrated that suppressing unsafe behavior can be effective in improving safety performance [34,35,36], and social opinions can serve as effective external factors to influence safety performances.

(2) Emotional orientation:

It is infrequent for construction projects to receive such widespread attention. Safety management boils down to promoting synergy [37]. Relationships between people change over the course of the collaboration, but perceptions generated by initial contacts have huge impacts on the outcome [38]. This paper divides management into three categories (investor, constructor, and labor subcontractor) and further connects them in a refined analysis, expanding into the relationship between investor and constructor, connection with the constructor, and fitting between first-line managers and labor subcontractors. The investor, as the top leader, has great authority ($B_{6}$), but because the investor’s main business is generally not construction, the investor’s construction expertise is weaker. The investor easily forgoes dictating the building site restrictions [39]. As the leader of the construction process, the project manager’s emotional intelligence ($B_{7}$) is a guarantee of the coordinated operation involving all departments [40]. The ambiguous intimacy ($B_{8}$) of the leadership is the root cause of the leader–member ambivalence; subordinates do not adequately have a united front with the leader and tend to take conservative or even stagnant approaches to performing their tasks [4]. Dechawatanapaisal [41] proposed that employees’ ambivalence has a direct impact on career commitment. Wu et al. [42] revealed that employees tend to adopt compromised approaches whether actively coping or passively avoiding such ambiguous intimacies. Ambiguous intimacy imposes a huge psychological burden on employees, making them walk on thin ice in work processes [43]. Employees’ perceived organizational statuses ($B_{9}$) [44] and proactive behaviors ($B_{10}$) [45] are two complementary factors influenced by overall security climate, representing employees’ perceived statuses and involvement in the building organizations as a whole.

(3) Stability factors:

The cornerstone of stable organizational relations lies in management mechanisms ($B_{17}$) [46]. Management mechanisms represent the resistance of the construction project to internal and external shocks. There are many factors to consider when carrying out a construction project, such as geological environment, risk sources, and human relations. The management system of a construction site can clarify the responsibilities of the project’s employees, and the safety culture ($B_{15}$) ($B_{18}$) of a company can influence the work engagement of the employees ($B_{14}$) [47]. With the gradual improvement of the informationization of modern construction projects, the role of intelligence has also come to the fore ($B_{16}$) [48,49]. Similar to the influences of social opinions of employees, the use of intelligence promotes transparency and informatization on construction sites, where employees’ unsafe behaviors can be monitored and stopped in a timely manner, which could increase safety awareness.

### 2.3. Cloud Model

Ambiguity and randomness are reflected in all aspects of human history, and the uncertainties they represent make it impossible to describe things precisely. For this reason, in 1995, Li Deyi, a member of the Chinese Academy of Engineering, proposed a concept of a cloud model to describe mathematically objective laws of this uncertainty phenomenon [50]. The pervasiveness of the cloud model has been widely demonstrated and has been extended to many domains [51,52,53]. In the field of construction engineering, cloud models were mainly used in safety risk evaluations, such as in the evaluation of construction site safety system resilience assessments [54] and comprehensive safety evaluations of building construction sites [55].

The cloud represents conversion and uncertainty mapping between qualitative and quantitative factors. The numerical characteristics of the cloud mainly consist of the expectation value $E_{x}$, entropy value $E_{n}$, and hyper-entropy $H_{e}$. $E_{x}$ is a central value of the cloud, which indicates the grade interval or average value of cloud droplets; the safety performances of construction projects under the influence of psychological states are based on this value; the entropy value $E_{n}$ reflects the sizes of the span of clouds and discrete degrees of cloud drops, a larger $E_{n}$ indicates greater randomness of the evaluation index and greater ambiguity of the grading boundary; hyper-entropy $H_{e}$ is an entropy uncertainty measure, reflecting the thickness of the cloud layer. In order to reduce various ambiguities and uncertainties in the process of the safety evaluation under the influence of the mental state, the cloud correlation model was constructed by using the expectation value $E_{x}$, entropy value $E_{n}$, and hyper-entropy $H_{e}$.

In 1983, Professor Cai Wen proposed topology, also known as matter–element analysis, as a system of methods to reveal the inner laws of contradictory problems and to solve contradictory problems under restrictive conditions [56]. The basic unit of topology is matter–element, and the basic model of matter–element is an ordered triple *R* = (name of problem, characteristic of problem, value taken of the characteristic) = (*N*, *C*, *V*). *N*, *C*, and *V* are three elements of matter–element, and *V* is usually a definite value or interval. The eigenvalue *V* of the matter–element analysis model is usually a definite value. By combining the cloud model with the matter–element analysis and taking advantage of fuzziness and randomness of the cloud model, ($E_{x}$, $E_{n}$, $H_{e}$) is used instead of *V* to construct a topological cloud model for the safety performances of construction projects under the influences of psychological states; these were qualitatively and quantitatively synthesized and evaluated, and uncertainties of the evaluations were considered. The specific mathematical model is shown below.

If there are *m* evaluation indicators for the matter–element to be evaluated, i.e., $C_{1}$, $C_{2}$, ⋯$C_{m}$, and each indicator has *n* evaluation levels, then the *m*-dimensional matter–element is:
(1)$$R=\left[\begin{array}{ccc} N & C_{1} & V_{1}\\ \; & C_{2} & V_{2}\\ \; & \vdots & \vdots \\ \; & C_{m} & V_{m} \end{array}\right]=\left[\begin{array}{c} R_{1}\\ R_{2}\\ \vdots \\ R_{m} \end{array}\right]$$

The matter–element extension cloud model can be expressed as: (2)$$R=\left[\begin{array}{ccc} N & C_{1} & \left(Ex_{1},En_{1},He_{1}\right)\\ N & C_{2} & \left(Ex_{2},En_{2},He_{2}\right)\\ \vdots & \vdots & \vdots \\ N & C_{n} & \left(Ex_{n},En_{n},He_{n}\right) \end{array}\right]$$

According to a large number of existing research results on construction project safety evaluations [54,55], relevant laws and regulations, as well as discussion results from experts, this paper divides the safety performances of construction projects under the influences of psychological states into three levels and determines the evaluation criteria of each level within the index system. Since the largest number of small- and medium-sized construction projects exist in China, when universally considering, an upper limit is set for the investment ($B_{1}$), area ($B_{2}$), and number of participants ($B_{3}$), in addition to referring to the relevant Chinese industry standards. There are many qualitative indicators in this system, and these indicators are evaluated in two ways in this paper—grade evaluation and score evaluation. The grade is divided into 1–5 levels; a higher grade means a stronger degree of performance of the index content; the score range is [0,100]. Numerical characteristics of the cloud are key to evaluate the model. Each indicator of the classification level has a value interval ($X_{\max}$,$X_{\min}$), and three elements of the standard cloud model (expected value, entropy, and hyper-entropy) are calculated using Equations (3)–(5). According to the evaluation method of the standard cloud model, the extension cloud matter–element model can be further constructed. Among them, *s* is denoted as a constant, which is set by the uncertainty of each indicator and actual situation; in this paper, considering the characteristics of each indicator, the constant value of *s* is set to 0.1 [55]. The cloud diagram is shown in Figure 3. The specific numerical divisions are shown in Table 3.
(3)$$E_{x}=\frac{X_{\max}+X_{\min}}{2}$$
(4)$$E_{n}=\frac{X_{\max}-X_{\min}}{6}$$
(5)$$H_{e}=s$$

### 2.4. Assignment of Subjective Weights

The analytic hierarchy process (AHP) is a decision-making method that decomposes elements that are relevant for decision-making into, e.g., objectives, criteria, and solutions, where qualitative and quantitative analyses are performed [57]. The subjective weights of the indicators are calculated by analyzing and comparing each indicator based on expert experience. The calculation steps of the analytic hierarchy process are as follows:

Step 1: Decompose the hierarchy of the target problem and construct the judgment matrix within each level. The judgment matrix of each level is constructed according to the “1 to 9” ratio scale proposed in the literature [58]. The results are expressed using $M_{ij}$, which represents the comparison between factor *i* and factor *j* in terms of importance, expressed in a formula as the following: $M_{ij}=1/M_{ji}$. The quantitative representation of the importance is shown in Table 4.

Step 2: Calculate the maximum eigenvalue ($\lambda_{\max}$) of the matrix. The consistency test is performed by following equations. The consistency of the matrix is checked by the consistency ratio (CR) to see if the consistency of the matrix meets the requirements. When CR<0.1, the consistency of the pairwise comparison matrix is judged to pass the requirements. The values of the average random consistency index RI are shown in Table 5.
(6)$$CI=\frac{\lambda_{\max}-n}{n-1}$$
(7)$$CR=\frac{CI}{RI}$$

The above steps allow subjectivity calculating the weights of the safety performances of the construction projects under the influences of psychological states (for each indicator), as shown in Table 6.

### 2.5. Assignment of Dynamic Weights

Dynamic weights are assigned by the position of the indicator’s magnitude in the classical domain, with the data closer to the center having more weight. The closer the value of the indicator is to the center of the range of values, the more standard the level represented by the indicator. The higher the value of the indicator in the range of the danger level, the more dangerous the level represented by the indicator, and the higher the weight assigned to it. Based on the literature [59], steps for implementing dynamic weighting are shown below:

Assume that the range of values of indicator $C_{i}$ in the risk level *j* is $v_{ij}=\left[a_{ij},b_{ij}\right]$.
(8)$$r_{ij}\left(\nu_{i},V_{ij}\right)=\langle \begin{array}{c} \frac{2(v_{i}-a_{ij})}{b_{ij}-a_{ij}},v_{i}\leq \frac{\left(a_{ij}+b_{ij}\right)}{2}\\ \frac{2(b_{ij}-v_{i})}{b_{ij}-a_{ij}},v_{i}\geq \frac{\left(a_{ij}+b_{ij}\right)}{2} \end{array},\left(i=1,2,\ldots ,n;j=1,2,\ldots ,m\right)$$
(9)$$r_{ij\max}\left(v_{i},V_{ij\max}\right)=\max_{j}\left(r_{ij}\left(v_{i},V_{ij}\right)\right)$$
(10)$$r_{i}=\langle \begin{array}{c} j_{\max}\cdot \left(1+r_{ij\max}\left(v_{i},V_{ij\max}\right)\right),r_{ij\max}\left(v_{i},V_{ij\max}\right)\geq -0.5\\ j_{\max}\cdot 0.5,r_{ij\max}\left(v_{i},V_{ij\max}\right)\leq -0.5 \end{array}$$
(11)$$\alpha_{i}=\frac{r_{i}}{\sum_{i=1}^{n}r_{i}}$$

### 2.6. Combined Weighting Calculation

In order to avoid the single-weight evaluation method from being too different from the actual situation, the game theory method is introduced to combine multiple weighting methods into a more reasonable weighting model. The implementation steps for combining weights are shown below:

Step 1: It is assumed that the model uses an *x*-group weighting approach (two in this paper), with each group of weight vectors denoted as $W_{k}=\left\{W_{k1},W_{k2},\ldots ,W_{kn}\right\}\left(k=1,2,\ldots ,x\right)$, and linear combination coefficients as $\alpha_{k}$. The combination is performed according to Equation (Equation 12): (12)$$W=\sum_{k=1}^{x}\alpha_{k}W_{k}^{T},\left(\alpha_{k}>0\right)$$

Step 2: According to the optimal strategy, minimize the deviation between *W* and $W_{k}$ to find the smallest $\alpha_{k}$, as shown in Equation (Equation 13): (13)$$\min{∥\sum_{k=1}^{x}\alpha_{k}W_{k}^{T}-W_{k}^{T}∥}^{2}\left(k=1,2,\cdots ,x\right)$$

The conditions for obtaining the optimal first-order derivative after the derivative of the matrix are given in Equation (Equation 14): (14)$$\sum_{j=1}^{x}\alpha_{j}W_{k}W_{j}^{T}=W_{k}W_{k}^{T};\left(k=1,2,\cdots ,x\right)$$

Step 3: Normalizing according to Equation (Equation 15), combined weights can be obtained by Equation (Equation 16): (15)$$\alpha_{k}^{*}=\alpha_{k}/\sum_{k=1}^{x}\alpha_{k}$$
(16)$$W^{*}=\sum_{k=1}^{x}\alpha_{k}^{*}W_{k}^{T}$$

### 2.7. Safety Level Measurement

The cloud model consists of cloud droplets, and the security level of the cloud model is determined according to the maximum affiliation principle. The affiliation degree of cloud droplet *x* in the standard cloud model is calculated by Equation (Equation 17): (17)$$k_{ij}=\exp\left(-\frac{{\left(x_{i}-E_{x}\right)}^{2}}{2{\left(E_{n}^{\prime }\right)}^{2}}\right)$$

The correlation matrix is denoted as *D*, where i=1,2,3,⋯,m;j=1,2,3, and $k_{ij}$ are the correlations between indicator *i* and level *j*.
(18)$$D=\left[\begin{array}{ccc} k_{11} & k_{12} & k_{13}\\ k_{21} & k_{22} & k_{23}\\ \vdots & \vdots & \vdots \\ k_{m1} & k_{m2} & k_{m3} \end{array}\right]$$

The combined evaluation vector *B* can be obtained by combining the weight vector with the correlation matrix: (19)$$B=WD=\left[b_{1},b_{2},b_{3}\right]$$

The value of the composite judging index *r* can be obtained by the weighted average method: (20)$$r=\frac{\sum_{i=1}^{3}b_{i}f_{i}}{\sum_{i=1}^{3}b_{i}}$$
where $b_{i}$ is a component of vector *B*. $f_{i}$ is the value of the evaluation level (1: low; 2: medium; 3: high).

Since $E_{n}^{\prime }$ is randomly generated when solving the cloud correlation degree *k*, and $x_{i}$ is also random in nature, multiple solving is required to reduce the effect of randomness on the evaluation results. In this paper, t=100.
(21)$$E_{rx}=\frac{\sum_{t=1}^{t}r_{i}\left(x\right)}{t}$$
(22)$$E_{rn}=\sqrt{\frac{\sum_{t=1}^{t}{\left(r_{i}\left(x\right)-E_{rx}\right)}^{2}}{t}}$$

The credibility θ is inversely proportional to the confidence of results, with larger θ and lower confidence.
(23)$$\theta =\frac{E_{rn}}{E_{rx}}$$

## 3. Example Analysis

### 3.1. Example 1

The Yueyang Workers’ Cultural Palace and Dongfeng Square were transformed into a comprehensive service place, integrating leisure, entertainment, culture, education, sports, fitness, and business offices. The construction includes a workers’ cultural palace, a standard sports field, two underground parking lots, roads, squares, surface parking spaces, greening, an outdoor pipeline network, and other supporting facilities. Through on-site research and expert discussions, engineering evaluation volume values were determined, and data of each index are shown in Table 7. The surrounding environment and planning of the project are shown in Figure 4.

(1) The index weights were assigned to the Yueyang City engineering project as an example. The subjective weights were assigned using hierarchical analysis, as shown in Table 6. Dynamic weight analysis was calculated by the cloud model correlation function with Equations (8)–(10). Using game theory combined with two weighting methods, the set of solution equations for optimal combination coefficients was established by Equations (12)–(16), i.e., Equation (Equation 24). Solving the equation shows that $\alpha_{1}=0.879$, $\alpha_{2}=0.272$. By normalizing it, $\alpha_{1}=0.764,\alpha_{2}=0.236$. The final weight assignments are shown in Table 8.
(24)$$\langle \begin{array}{c} \alpha_{1}W_{1}{W_{1}}^{T}+\alpha_{2}W_{1}{W_{2}}^{T}=W_{1}W_{1}^{T}\\ \alpha_{1}W_{2}{W_{1}}^{T}+\alpha_{2}W_{2}{W_{2}}^{T}=W_{2}W_{2}^{T} \end{array}$$

(2) The second step is to use the cloud model correlation function to calculate the correlation degree of each level of the indicator. The calculation results are shown in Table 9, and the risk level of each indicator of the project is shown in Figure 5.

(3) From Equations (19)–(23), the safety level and trustworthiness can be calculated. $E_{rx}=1.805686827,\theta =0.015504463$, i.e., the selected project is at a medium risk level.

### 3.2. Example 2

The project is located in the Shangrao high-speed railway economic development zone. The project is surrounded by the city’s main roads (Tianyou Avenue, Cha Sheng Road, and Xinyuan Road), is a very advantageous location, and is the center of urban cultural activities. The whole block is diamond-shaped, with a total land area of about 101,012 square meters (about 151 mu). The existing vegetation is good and can be preserved in principle. The existing strong topography is lower, flat, with local hilly terrain, and average in height; the difference is about 15.0 m. Construction works include Guang Sheng Temple, Guang Sheng Pagoda, She Gong Temple, ancillary rooms, tea houses, management rooms, corridors, etc.; environmental works include ecological green spaces, paved squares, scenic walls, bar streets, natural water features, air stacks, etc. Since this project is a municipal project, which covers a large but not very difficult area, only the area of the housing project was considered. The specific data are shown in Table 10.

Calculated by a formula similar to that of Example 1, $\alpha_{1}=0.866,\alpha_{2}=0.134$. The weights are assigned as shown in Table 11. The combined correlations of indicators are shown in Table 12. The risk level of each indicator is shown in Figure 6. The calculation of the safety level is similar to that of Example 1. $E_{rx}=1.8481547111,\theta =0.019848204$, i.e., selected project is at a medium risk level.

## 4. Discussion

The assessment and analysis of the psychological states of personnel involved in construction projects are critical for maintaining excellent safety performances on projects. Due to the complex management objectives, interest orientations of construction site management, ambiguous cooperation directions, and elusive superior–subordinate relationships, research is limited concerning personnel relationships and the behavioral orientations under the influences of psychological factors. Therefore, this paper used an extension cloud model to construct the safety performances of construction projects under the influence of the psychological state evaluation model.

By decomposing the target of safety performances under the influences of psychological states, 3 secondary indicators, and 18 specific indicators were obtained to build a perfect evaluation system of safety performance levels under the influences of psychological states in construction projects. The system was used in two specific construction projects to evaluate and analyze safety performances under the influences of psychological states. In addition, this paper introduced an extension cloud model based on this evaluation system, taking into account uncertainty and fuzziness, so that evaluation results are closer to actual situations. Meanwhile, to prevent the weighting method from deviating from the actual situation, the game theory combined weighting method was used to combine subjective weights with dynamic weights.

The final risk rating of both examples is medium. Example 1 consisted of a large investment, located in an urban area; it has a complex surrounding environment and a large number of participants, so it can represent a complex project. Example 2 consisted of a low investment, located in a suburban area, and covering a vast area but with low operational difficulty. It has a simple surrounding environment that receives less attention from the public, so it can represent an ordinary project. Both construction projects were completed on time and in quality, and no safety accidents or personnel disputes occurred during the construction process. The assessment conclusions in this paper are consistent with the actual project operation results.

From the evaluations of the index levels of two specific projects, it can be seen that, except for the safety levels of the fixed factors, such as the number of participants, total investment amount, and floor space, which cannot be manipulated artificially, most of the other qualitative indicators can be reduced to risk levels by management or subjective regulations. In complex projects, projects receive strong attention from the public, the participant numbers are large and participating units are mostly large enterprises. Management culture and project systems at each management level are also more perfect; each branch of the project has a corresponding management system of supervisors, and project risks mainly exist in mutual influences of personnel relations. Due to the ambiguity of management’s hierarchy and intersections of managing branches, the functions of managers may overlap, but their respective interests are not equally oriented, making them prone to conflict, resulting in a decrease in safety performances [60]. Conflict will only hinder the proper operations of the project, and good teamwork will bring a project to success. In ordinary projects, most indicator safety levels are at medium and low levels. Small- and medium-sized construction projects receive less attention from public opinion, and governments and the public do not monitor these projects as much as they do with complex projects. The relationships between personnel within projects will become close and harmonious in a relaxed atmosphere, and because the risk sources of small- and medium-sized construction projects are fewer and weaker, it is difficult for large safety accidents to occur. Most participating units are also small- and medium-sized enterprises, and their management culture and management systems are slightly inadequate compared with those of large enterprises, resulting in weaker attention to safety atmospheres by managers in such projects [61]. While human factors are fundamental to maintaining project safety performances, they are also influenced by many external factors. Excellent safety culture and management mechanisms can lead to good people connections and synergy.

Therefore, the following recommendations are given in this paper:

(1) Many studies have shown that the majority of safety incidents on construction sites are related to human factors. This paper presents superior–subordinate conflicts as important points in human factors. The comparative analyses of complex projects and ordinary projects show that excellent management systems and safety culture factors can restrain the unconscious generation of hostility and disagreement in human interactions. Simultaneously, safety climate factors, which are considered to be the employees’ guiding beliefs on safety issues, also affect the safety performances of projects [62]. Proactive safety with employees can have positive effects on promoting safety performances [63]. Therefore, it is necessary to improve safety systems at construction sites and strengthen safety culture, which will have positive effects in restraining the bad behaviors of employees and in fostering positive safety climates in construction sites.

(2) In addition to human factors, intelligence and standardization of building sites are equally important. Intelligent building sites proposed in this paper have become a trend in China’s construction industry. China’s epidemic management measures require that each building site establish good intelligent supervision measures; excellent results have shown that these measures are worth promoting in other aspects of building sites, especially in safety management.

(3) The development of the construction industry is a cornerstone of a country’s development; construction projects also receive public attention, with construction processes affecting every aspect of the public’s life. The extent to which a construction project receives public attention can greatly affect the attitude of staff members. In the case of the Leishenshan and Huoshenshan Hospital, for example, the construction process was broadcast live around the clock, and all of the project’s goals were accomplished excellently. Effective public opinion monitoring has a catalytic effect on the safety performances of construction projects.

## 5. Conclusions

This paper establishes a safety performance evaluation model for construction projects under the influence of a psychological state based on the extension cloud model, and it verifies the applicability of the evaluation system and evaluation model established in this paper by comparing the evaluation analyses of actual cases with the actual operation results of projects. Through a comparative analysis of a complex construction project and an ordinary construction project, based on the safety evaluation system proposed in this paper, it can be seen that objective factors, such as the number of participants, total investment amount, and floor space are the roots of a project’s risk level, and they cannot be changed. By refining emotion-oriented indicators, some external manifestations are used to describe the psychological impacts of the abstract ambiguous intimacies of employees. In complex projects, excellent safety systems and sound site facilities can offset some personnel conflicts, but may also inspire negative attitudes. In ordinary projects, a relaxed working environment can lead to close personnel relations, but it can also create a lack of an objective environment and cause safety hazards. The analysis shows that the model proposed in this paper can be used for the evaluation of safety performances for real construction projects, and it provides help for project managers to grasp overall safety performance levels.

With advances in technology, changes in national policies, and the long-term ongoing nature of construction projects, internal management mechanisms of a building site are not static. It is impractical to completely dissect the impacts of building site management interactions on the safety performances through a model. We used an evaluation system to find the key factors that influence the relationships between people and suggest improvements to improve human-centered safety performance measures.

In future work, we will extend the study to observe how more factors influence the safety performances of a building site, in terms of human, physical, and environmental factors. A quality mathematical model was used to analyze operational paths to determine excellent safety performances and provide more clarity on the requirements and conditions needed to achieve them.

## Figures and Tables

**Figure 1 ijerph-19-15378-f001:**
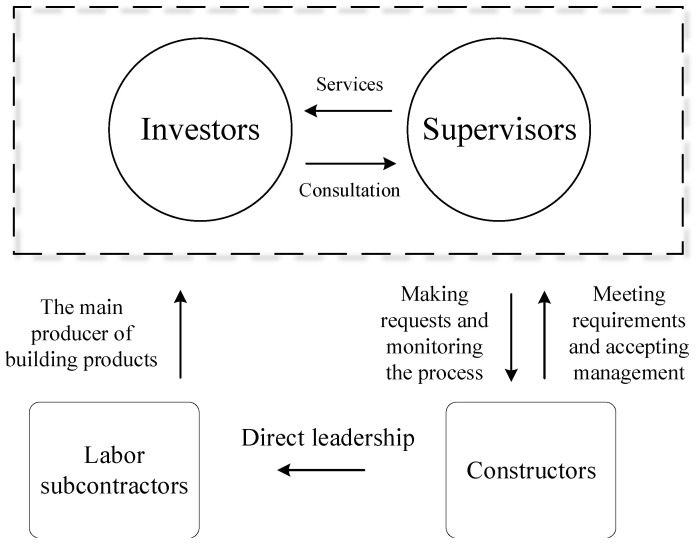
Tripartite relationship of investors–constructors–labor constructors.

**Figure 2 ijerph-19-15378-f002:**
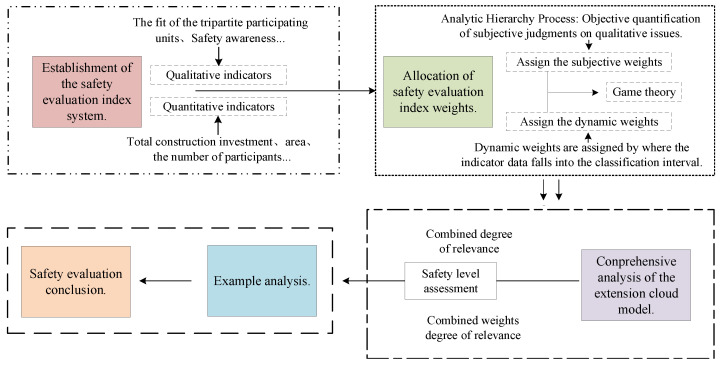
Research method framework.

**Figure 3 ijerph-19-15378-f003:**
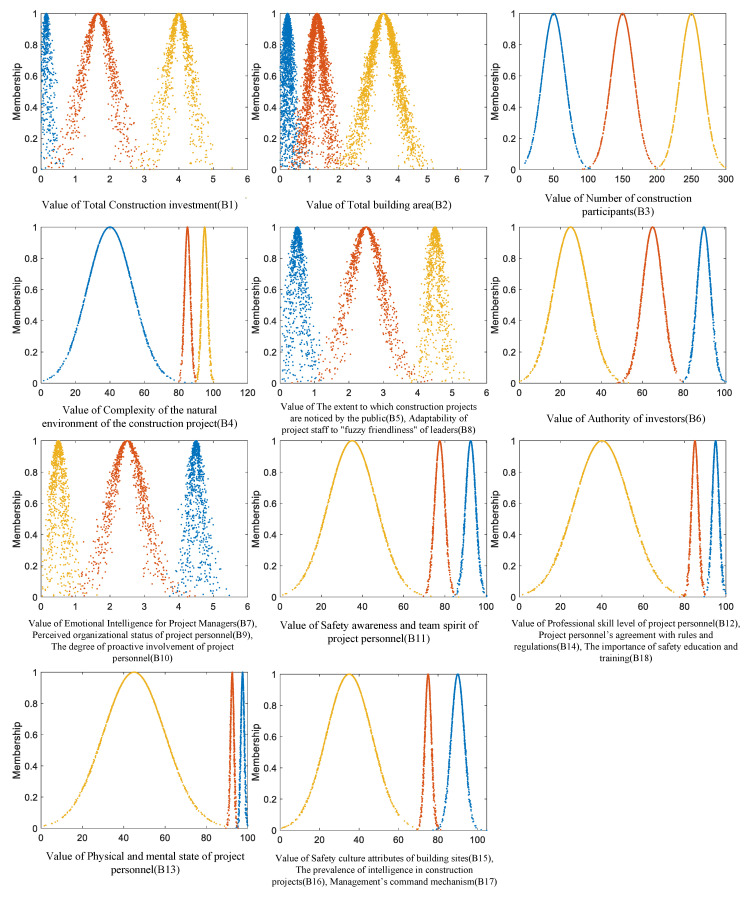
Cloud diagram of the evaluations of the safety performances of construction projects under the influences of psychological states.

**Figure 4 ijerph-19-15378-f004:**
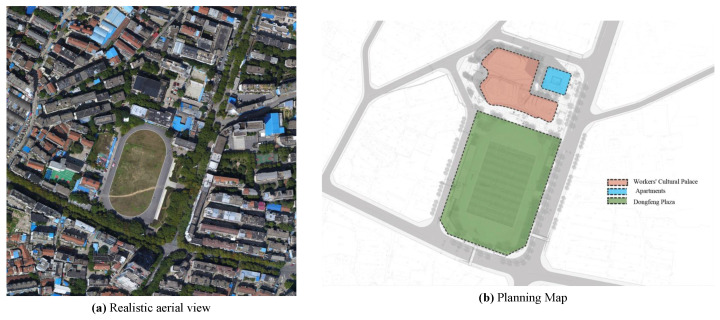
Relevant real-world view of the project.

**Figure 5 ijerph-19-15378-f005:**
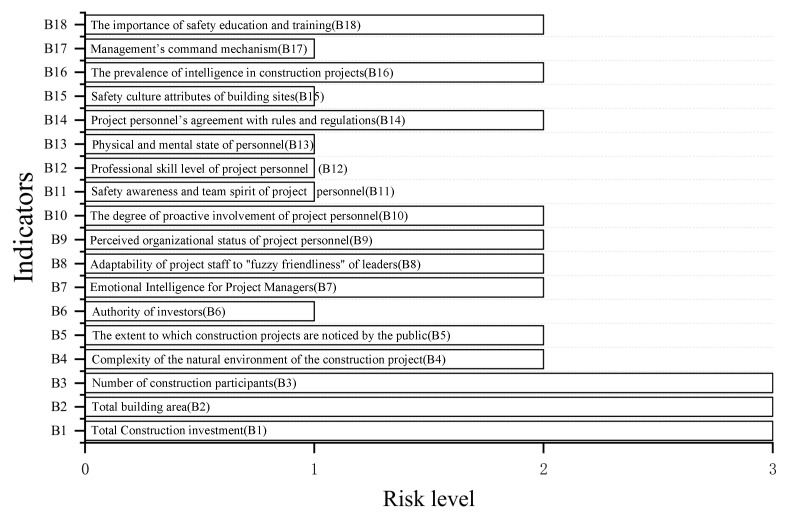
Risk level of each indicator.

**Figure 6 ijerph-19-15378-f006:**
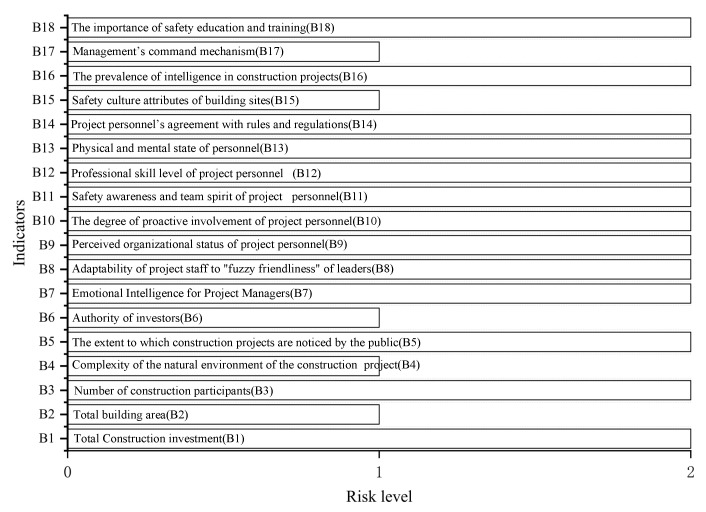
Risk level of each indicator.

**Table 1 ijerph-19-15378-t001:** Demographic information of the participants

Expert	Type of Unit	Career	Education	Age
Project manager 1	Construction Company	Construction Project Manager/Senior Engineer	Bachelor degree	50
Project manager 2	Construction Company	Construction Project Manager/Senior Engineer	Bachelor degree	38
Senior engineer 1	Engineering Design Institute	Construction Project Manager/Senior Engineer	Master degree	44
Senior engineer 2	Engineering consulting firm	Consulting Engineer	Master degree	38
Senior engineer 3	Construction Company	Construction Project Manager/Senior Engineer	junior college	36
Expert 1	Scientific Research Institutes	University Professor (Construction Management)	Doctor degree	49
Expert 2	Scientific Research Institutes	University Professor (Construction Management)	Doctor degree	52
construction technician 1	Construction Company	Construction Worker	junior college	25
construction technician 2	Construction Company	Construction Worker	Bachelor degree	24
construction technician 3	Construction Company	Construction Worker	Bachelor degree	24

**Table 2 ijerph-19-15378-t002:** Indicator system for the safety performances of construction projects under the influence of psychological states.

Target	Second Indicators	Specific Indicators	Scale Range
Safety performance of construction projects under the influence of psychological states	Building scale ($A_{1}$)	Total construction investment ($B_{1}$)	Billion:[0,5)
		Total building area ($B_{2}$)	10,000 Square meters:[0,5)
		Number of construction participants ($B_{3}$)	Number of people:[0,300)
		Complexity of the natural environment of the construction project ($B_{4}$)	Points:[0,100)
		The extent to which construction projects are noticed by the public ($B_{5}$)	Level:{0,1,2,3,4,5}
	Emotional orientation ($A_{2}$)	Authority of investors ($B_{6}$)	Points:[0,100)
		Emotional Intelligence for Project Managers ($B_{7}$)	Level:{0,1,2,3,4,5}
		Adaptability of project staff to “fuzzy friendliness” of leaders ($B_{8}$)	Level:{0,1,2,3,4,5}
		Perceived organizational status of project personnel ($B_{9}$)	Level:{0,1,2,3,4,5}
		The degree of proactive involvement of project personnel ($B_{10}$)	Level:{0,1,2,3,4,5}
	Stability factors ($A_{3}$)	Safety awareness and team spirit of project personnel ($B_{11}$)	Points:[0,100)
		Professional skill level of project personnel ($B_{12}$)	Points:[0,100)
		Physical and mental state of project personnel ($B_{13}$)	Points:[0,100)
		Project personnel’s agreement with rules and regulations ($B_{14}$)	Points:[0,100)
		Safety culture attributes of building sites ($B_{15}$)	Points:[0,100)
		The prevalence of intelligence in construction projects ($B_{16}$)	Points:[0,100)
		Management’s command mechanism ($B_{17}$)	Points:[0,100)
		The importance of safety education and training ($B_{18}$)	Points:[0,100)

**Table 3 ijerph-19-15378-t003:** Criteria for evaluating the safety performances of construction projects under influences of psychological states and cloud model boundaries.

Indicator	Evaluation Criteria			Cloud Model Boundaries		
	Low	Medium	High	Low	Medium	High
$B_{1}$ (billion)	[0,0.3)	[0.3,3)	[3,5)	(0.15,0.05,0.1)	(1.65,0.45,0.1)	(4,0.333,0.1)
$B_{2}$ (10,000 sqm)	[0,0.5)	[0.5,2)	[2,5)	(0.25,0.083,0.1)	(1.25,0.25,0.1)	(3.5,0.5,0.1)
$B_{3}$	[0,100)	[100,200)	[200,300)	(50,16.667,0.1)	(150,16.667,0.1)	(250,16.667,0.1)
$B_{4}$ (points)	[0,80)	[80,90)	[90,100)	(40,13.333,0.1)	(85,1.667,0.1)	(95,1.667,0.1)
$B_{5}$ (level)	[0,1)	[1,4)	[4,5)	(0.5,0.167,0.1)	(2.5,0.5,0.1)	(4.5,0.167,0.1)
$B_{6}$ (points)	(100,80]	(80,50]	(50,0]	(90,3.333,0.1)	(65,5,0.1)	(25,8.333,0.1)
$B_{7}$ (level)	(5,4]	(4,1]	(1,0]	(4.5,0.167,0.1)	(2.5,0.5,0.1)	(0.5,0.167,0.1)
$B_{8}$ (level)	[0,1)	[1,4)	[4,5)	(0.5,0.167,0.1)	(2.5,0.5,0.1)	(4.5,0.167,0.1)
$B_{9}$,$B_{10}$ (level)	(5,4]	(4,1]	(1,0]	(4.5,0.167,0.1)	(2.5,0.5,0.1)	(0.5,0.167,0.1)
$B_{11}$ (points)	(100,85]	(85,70]	(70,0]	(92.5,2.5,0.1)	(77.5,2.5,0.1)	(35,11.667,0.1)
$B_{12}$ (points)	(100,90]	(90,80]	(80,0]	(95,1.667,0.1)	(85,1.667,0.1)	(40,13.333,0.1)
$B_{13}$ (points)	(100,95]	(95,90]	(90,0]	(97.5,0.833,0.1)	(92.5,0.833,0.1)	(45,15,0.1)
$B_{14}$ (points)	(100,90]	(90,80]	(80,0]	(95,1.667,0.1)	(85,1.667,0.1)	(40,13.333,0.1)
$B_{15}$,$B_{16}$,$B_{17}$ (points)	(100,80]	(80,70]	(70,0]	(90,3.333,0.1)	(75,1.667,0.1)	(35,11.667,0.1)
$B_{18}$ (points)	(100,90]	(90,80]	(80,0]	(95,1.667,0.1)	(85,1.667,0.1)	(40,13.333,0.1)

**Table 4 ijerph-19-15378-t004:** Numerical quantification of the hierarchical analysis method.

Quantified Values	Relative Degree	Quantified Values	Relative Degree
1	Equally important	7	Intensely important
3	Slightly more important	9	Extremely important
5	Stronger and more important	2, 4, 6, 8	Intermediate

**Table 5 ijerph-19-15378-t005:** Value of RI.

Order	1	2	3	4	5	6	7	8	9	10
RI	0	0	0.58	0.90	1.12	1.24	1.32	1.41	1.45	1.49

**Table 6 ijerph-19-15378-t006:** Weights of all indicators.

Second Indicators	Weight	Specific Indicators	Weight	Overall Weight
$A_{1}$	0.094	$B_{1}$	0.280	0.026
		$B_{2}$	0.122	0.011
		$B_{3}$	0.104	0.010
		$B_{4}$	0.230	0.022
		$B_{5}$	0.264	0.025
$A_{2}$	0.627	$B_{6}$	0.079	0.049
		$B_{7}$	0.092	0.058
		$B_{8}$	0.397	0.249
		$B_{9}$	0.240	0.150
		$B_{10}$	0.192	0.120
$A_{3}$	0.279	$B_{11}$	0.150	0.042
		$B_{12}$	0.071	0.020
		$B_{13}$	0.172	0.048
		$B_{14}$	0.034	0.009
		$B_{15}$	0.140	0.039
		$B_{16}$	0.093	0.026
		$B_{17}$	0.081	0.023
		$B_{18}$	0.259	0.072

**Table 7 ijerph-19-15378-t007:** Measurements of the indicators for the Yueyang project.

**Indicator**	$B_{1}$	$B_{2}$	$B_{3}$	$B_{4}$	$B_{5}$	$B_{6}$
**Value**	3.5	3.8	278	86	3	85
**Indicator**	$B_{7}$	$B_{8}$	$B_{9}$	$B_{10}$	$B_{11}$	$B_{12}$
**Value**	3.6	3	3.5	3.5	88	92
**Indicator**	$B_{13}$	$B_{14}$	$B_{15}$	$B_{16}$	$B_{17}$	$B_{18}$
**Value**	97	87	90	75	90	89

**Table 8 ijerph-19-15378-t008:** Final weight assignments.

**Indicator**	$B_{1}$	$B_{2}$	$B_{3}$	$B_{4}$	$B_{5}$	$B_{6}$
**Dynamic weights**	0.086	0.104	0.083	0.069	0.064	0.028
**Combined Weights**	0.04	0.033	0.027	0.033	0.034	0.044
**Indicator**	$B_{7}$	$B_{8}$	$B_{9}$	$B_{10}$	$B_{11}$	$B_{12}$
**Dynamic weights**	0.049	0.064	0.051	0.051	0.027	0.027
**Combined Weights**	0.056	0.205	0.127	0.104	0.038	0.022
**Indicator**	$B_{13}$	$B_{14}$	$B_{15}$	$B_{16}$	$B_{17}$	$B_{18}$
**Dynamic weights**	0.035	0.061	0.038	0.077	0.038	0.046
**Combined Weights**	0.045	0.021	0.039	0.039	0.027	0.066

**Table 9 ijerph-19-15378-t009:** Comprehensive correlation.

Indicator	Low	Medium	High	Max
$B_{1}$	0	0	0.532	0.532
$B_{2}$	0	0	0.864	0.864
$B_{3}$	0	0	0.244	0.244
$B_{4}$	0.002	0.821	0	0.821
$B_{5}$	0	0.685	0	0.685
$B_{6}$	0.335	0	0	0.335
$B_{7}$	0.014	0.115	0	0.115
$B_{8}$	0	0.492	0	0.492
$B_{9}$	0	0.110	0	0.110
$B_{10}$	0	0.159	0	0.159
$B_{11}$	0.2	0	0	0.2
$B_{12}$	0.173	0	0	0.173
$B_{13}$	0.814	0	0.02	0.814
$B_{14}$	0	0.517	0.02	0.517
$B_{15}$	1	0	0	1
$B_{16}$	0	1	0.003	1
$B_{17}$	1	0	0	1
$B_{18}$	0	0.124	0.01	0.124

**Table 10 ijerph-19-15378-t010:** Measurements of indicators for the Shangrao project.

**Indicator**	$B_{1}$	$B_{2}$	$B_{3}$	$B_{4}$	$B_{5}$	$B_{6}$
**Value**	0.9	0.323	110	70	2	85
**Indicator**	$B_{7}$	$B_{8}$	$B_{9}$	$B_{10}$	$B_{11}$	$B_{12}$
**Value**	3.2	3.8	3.5	3	75	88
**Indicator**	$B_{13}$	$B_{14}$	$B_{15}$	$B_{16}$	$B_{17}$	$B_{18}$
**Value**	94	83	82	75	88	82

**Table 11 ijerph-19-15378-t011:** Final weight assignments.

**Indicator**	$B_{1}$	$B_{2}$	$B_{3}$	$B_{4}$	$B_{5}$	$B_{6}$
**Dynamic weights**	0.061	0.038	0.051	0.027	0.071	0.032
**Combined Weights**	0.031	0.015	0.015	0.023	0.031	0.047
**Indicator**	$B_{7}$	$B_{8}$	$B_{9}$	$B_{10}$	$B_{11}$	$B_{12}$
**Dynamic weights**	0.078	0.048	0.057	0.071	0.071	0.059
**Combined Weights**	0.061	0.222	0.138	0.113	0.046	0.025
**Indicator**	$B_{13}$	$B_{14}$	$B_{15}$	$B_{16}$	$B_{17}$	$B_{18}$
**Dynamic weights**	0.059	0.068	0.025	0.085	0.038	0.059
**Combined Weights**	0.049	0.017	0.037	0.034	0.025	0.07

**Table 12 ijerph-19-15378-t012:** Comprehensive correlation.

Indicator	Low	Medium	High	Max
$B_{1}$	0.001	0.243	0	0.243
$B_{2}$	0.627	0.014	0	0.627
$B_{3}$	0.002	0.056	0	0.056
$B_{4}$	0.079	0	0	0.079
$B_{5}$	0	0.589	0	0.589
$B_{6}$	0.322	0	0	0.322
$B_{7}$	0	0.372	0	0.372
$B_{8}$	0	0.042	0.016	0.042
$B_{9}$	0.002	0.146	0	0.146
$B_{10}$	0	0.588	0	0.588
$B_{11}$	0	0.604	0.003	0.604
$B_{12}$	0	0.196	0.002	0.196
$B_{13}$	0	0.203	0.005	0.203
$B_{14}$	0	0.487	0.006	0.487
$B_{15}$	0.057	0	0	0.057
$B_{16}$	0	1	0.003	1
$B_{17}$	0.835	0	0	0.835
$B_{18}$	0	0.199	0.007	0.199

## Data Availability

Not applicable.

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
