# Peer review of "Safety Performance Assessment of Construction Sites under the Influence of Psychological Factors: An Analysis Based on the Extension Cloud Model"

_ijerph, 2022, doi:10.3390/ijerph192215378_

Round 1

Reviewer 1 Report

Dear authors, 

please address my humble comments. 

kindest regards

Author Response

Dear Reviewer

Thank you very much for your careful reading and valuable suggestions. We have made corresponding changes based on your comments, please check them in the attachment.

Once again thank you for taking the time to review this paper during your busy schedule.

Sincerely

Junlong Peng and Qi Zhang

Reviewer 2 Report

Please check attached file for detail comments. 

Author Response

(The authors gave the same response as above.)

Reviewer 3 Report

Very interesting study, which approaches the construction safety topic from different perspective. Although the problem statement is clearly presented and justified, I would suggest to remove the generic statements about the construction industry and its safety performance, as many have reported same facts. 

The methodological approach, which is very appropriate, has been elaborated and clearly presented.

The authors have also discussed the research findings in detail by pulling out the significant outcomes. 

Overall, I would recommend the suitability of the manuscript for publication; however, the language needs to be improved. A professional proofreading is required for that.  

Author Response

Dear Reviewer

Thank you for your recognition of our work and very valuable comments. Following your comments, we have removed some redundant explanations. And we invited a native English teacher to check and polish our manuscript.

Once again thank you for taking the time to review this paper during your busy schedule.

Sincerely

Junlong Peng and Qi Zhang

Round 2

Reviewer 2 Report

The authors still need to address following concerns:

1. The authors still need to clarify how was the evaluation system developed.  For instance, how does Leader-Member Exchange Ambivalence guide the development of the evaluation system?   There is no Leader-Member Exchange in existing system.  If this is used to depict the interactions between different parts (e.g., investors and contractors), more relevant literature should be provided. 

2. Query in first review is not addressed: this paper extracts the factors of building scale, emotional orientation and stability as the secondary indicators of the evaluation system through industry norms, existing literature and expert discussion and so on.  However, it was not clear for the reviewer why and how these three aspects are related with psychological factors. How were the three perspectives selected and 18 factors determined? It seems no introduction in previous section.

3. The authors intend to study the influence of psychological status of the employees (?) from three parties on safety performance.  The parties include investors, contractors, and subcontractors.  The question is: 1) where is the data from, investor? Contractor? Or “and two project managers, three senior engineers, two experts engaged in research in this field, and three construction technicians?. If yes, how is the demographic information of the participants.  In what way they make assessment (questionnaire, interview or other means).

4. The authors add a section with title as “Influencing Factors of Safety Performance”, but the content should be improved.  The review of literature should be placed on the safety performance, and be fit with the proposed evaluation system.   

Author Response

(The authors gave the same response as above.)
